# Incorporating transportation costs into the single and multiple items newsvendor problems

Moncer Hariga ![ORCID] *

Department of Industrial Engineering, American University of Sharjah, Sharjah, United Arab Emirates

* mhariga@aus.edu

**Data Availability Statement:** All relevant data are within the paper and its Supporting information files.

**Funding:** This work was supported in part by the Open Access Program from the American

## Abstract

The efficiency and responsiveness of supply chains are vitally dependent on inventory replenishment and transportation decisions. In this paper, we study a supply chain consisting of a single retailer ordering seasonal products within the newsvendor framework. The primary objective of the paper is to investigate the retailer's decision-making process, aimed at determining the optimal replenishment quantities and selecting the appropriate mix and size of the truck fleet. Initially, we formulate a mathematical model where the retailer exclusively manages a limited fleet of its own trucks for inbound transportation of a single seasonal product. In this context, we determine a lower breakeven point for the fixed transportation cost than what has been previously proposed in the literature. Subsequently, we examine a commonly encountered transportation scenario where the retailer has the opportunity to expand its fleet size by leasing trucks from the external market. The outcomes of the numerical example indicates that the flexibility resulting from the utilization of different types of trucks can lead to reduced overall costs. We also address the practical transportation problem of efficiently shipping various seasonal products solely with the retailer's own trucks. For this complex problem, we propose an optimal solution procedure based on Lagrangian method. We show that the joint replenishment of multiple products results in cost savings and enhances utilization of the trucks' capacities.

## Introduction

Inventory replenishment and transportation decisions play a crucial role in enhancing the efficiency and responsiveness of supply chain operations. However, in practice, these critical decisions are taken sequentially and independently, leading to suboptimal supply chains performance [1]. Typically, the timing and quantity of the orders are established first. Subsequently, the number of trucks needed to transport the specified ordering quantity is determined, resulting in an inefficient utilization of the transportation resources. Therefore, it is essential that inventory and transportation decisions to be made concurrently rather than sequentially. This paper addresses the joint optimization of inventory and transportation problems within the newsvendor framework, for single and multiple items.

University of Sharjah. No additional external funding was received for this study.

**Competing interests:** The authors have declared that no competing interests exist.

The newsvendor problem is one of the classical and popular problems in stochastic inventory management [2]. It is concerned with the determination of the optimal ordering quantity for a seasonal product prior to the start of the season and before observing its demand. This classical problem has received growing attention from researchers and practitioners since the pioneering work by Arrow et al. [3]. In particular, numerous products experience short selling periods due to the rapid advancements of technology, which has led to shortened product life cycles, and the competitive pressures imposed by time-based competition. Consequently, the newsvendor model can accurately depict such real-life situations.

The retailer in the newsvendor problem incurs a unit purchasing cost, a per-unit shortage cost for unmet demand, and a per-unit holding cost for unsold inventory. In most of the newsvendor problems discussed in the literature, it is assumed that the per-unit transportation cost, often referred to as the less-than-truckload (LTL) assumption, is implicitly included in the unit purchasing costs. However, in this paper, transportation costs—both per-unit and per-truck—are not considered as part of the unit purchasing cost; instead, they are explicitly incorporated into the objective functions of the single and multiple item newsvendor problems. The relaxation of the LTL assumption enhances the realism of our proposed models and generate more efficient inventory and transportation policies.

For its inbound transportation, the retailer can exclusively rely on its own fleet of trucks, opts for leased trucks, or decide to completely outsource its transportation operations to third-party logistics providers. Furthermore, the retailer can consider shipping portion of the shipment quantity using its capacity-limited truck fleet and lease additional trucks when needed. The integration of these transportation options with inventory replenishment decisions can enhance the retailer's operational efficiencies. Indeed, leasing trucks can result in cost savings when ordering large quantities. Additionally, leasing provides the retailer access to additional trucks without incurring the substantial initial capital expenditure to acquire new trucks. Outsourcing transportation services provides numerous benefits, including economies of scale, capital investment savings, and reduced financial [4]. Likewise, outsourcing transportation operations can enhance efficiency, cost control, reliability, and delivery speed [5]. In this paper, we investigate the option of leasing trucks to transport a portion of the order quantity for the single item newsvendor problem.

The transportation of single items or multiple items depends on several factors such as the nature of the items, the selected transportation mode, and cost considerations. Transporting multiple items that share similar packaging characteristics, shapes, and weights, can result in cost savings and more efficient utilization of the truck's capacity. Indeed, aggregate shipping streamlines the processes of materials handling, packaging, and loading as it reduces the need for excessive handling and packaging and at the same time aids in the efficient utilization of available space within the trucks. This paper also addresses the muti-item newsvendor problem, taking transportation into consideration when developing the optimal ordering quantity for each item.

The rest of the paper is organized as follows. In the second section, we review papers in the literature that are relevant to the problems discussed in this paper. The third section presents the formulation and solution procedures for three integrated inventory and transportation problems within the newsvendor framework. Initially, we formulate a foundational model where a retailer is transporting a single product, utilizing solely its own fleet of trucks [6]. This establishes a baseline for understanding how inventory and transportation challenges are managed with limited logistics resources. In the second model, we expand the retailer's logistics resources by considering the option to lease trucks. This modification allows for a more flexible transportation strategy, accommodating scenarios where the retailer's own fleet may be insufficient due to capacity constraints. The third model further extends the operational scope

by relaxing the single-product assumption inherent in the first model. This model acknowledges the multifaceted nature of real-world supply chains, where retailers often need to balance multiple products with varying demand patterns and transportation requirements. Together, the three models offer a thorough examination of the integrated inventory and transportation problem, each introducing greater complexity and closer alignment with real-world scenarios. By progressively relaxing key assumptions and expanding the operational considerations, we provide enhanced insights on the ways retailers can refine their inventory management and transportation policies. The paper concludes with a final section that summarizes key findings and suggests potential avenues for future research.

## Literature review

Most of the deterministic and stochastic inventory models assumes LTL transportation mode, as they implicitly include the per-unit transportation cost into the purchasing cost. They further assume that the capacity of a single truck is sufficiently large to accommodate any ordering quantity. On the other hand, the full truck load (FTL) mode, where a fixed fee is charged per truck, is commonly utilized for transporting large quantities of goods. For a comprehensive discussion of various transportation modes, interested readers can refer to the review paper by Engebrethsen and Dauzère-Pérès [7].

The explicit incorporation of transportation costs into inventory models has a significant impact on the performance of supply chains (SC) operations. This has prompted SC researchers to study the implications of jointly optimizing inventory and transportation decisions. Yıldırmaz et al. [8] pointed out that omitting transportation costs can decrease the profit by an average of 2.25% and, in the worst cases, even up to 60%. Mendoza and Ventura [9] also concluded that neglecting transportation costs can increase monthly logistic cost by an average of 14.7% and transportation costs by 88.9%.

The effectiveness and efficiency of SCs can be affected by a variety of transportation challenges. These challenges encompass optimizing delivery routes, determining the number of trucks to deploy, and selecting the types and size of the truck fleet (Golden et al. [10]). The route optimization problem involves finding the most efficient delivery routes considering factors like distance travelled, traffic congestion, and the types of truck used. The feet size problem is concerned with the determination of the appropriate number of vehicles to be utilized to transport a specific load shipment. The joint selection of truck types and the fleet size is commonly referred to as the fleet mix and size problem. Each of these challenges has been extensively studied in transportation literature.

Lippman [11, 12], Iwaniec [13], and Aucamp [14] were among the first to consider FTL transportation costs in inventory models. Abdelwahab and Sargious [15] and Swenseth and Godfrey [16] compared the cost performance of FTL and LTL transportation modes when integrated one at a time with inventory decisions. Rieskts and Ventura [17] derived exact algorithms to generate optimal polices for inventory models using FTL and LTL transportation modes. They assumed constant demand over finite and infinite planning horizon. Li and Hai [1] investigated a joint inventory and transportation EOQ-based problem involving a truck capacity constraint. All of the aforementioned papers, which integrated inventory and transportation decisions, were developed within the EOQ framework, assuming deterministic demand.

Very few studies have jointly optimized inventory and transportation decisions in the context of stochastic demand. Zhang et al. [6] generalized the standard newsvendor model to include a per-truck transportation cost in the expected profit function. They showed that the optimal order quantity can be either the newsvendor solution or a multiple of the truck's

**Table 1. A classification of the relevant integrated inventory and transportation models.**

| Reference | SC Configuration | Demand | | Number of items | | Transportation cost structure | | Trucks ownership | |
|---|---|---|---|---|---|---|---|---|---|
| | | Deterministic | Stochastic | Single | Multiple | Fixed | Variable | Owned | Leased |
| [1] | Single stage | √ | | √ | | √ | √ | √ | |
| [2, 3] | Single stage | | √ | √ | | | √ | - | - |
| [6] | Single stage | | √ | √ | | √ | | √ | |
| [9] | Single stage | √ | | √ | | √ | √ | √ | |
| [11] to [17] | Single stage | √ | | √ | | √ | √ | √ | |
| [18] | Single stage | | √ | √ | | √ | √ | √ | |
| [19, 20] | Vendor-buyer | | √ | √ | | √ | √ | √ | |
| [21] | Single stage | | √ | | √ | √ | √ | √ | |
| This paper | Single stage | | √ | √ | √ | √ | √ | √ | √ |

capacity. Additionally, they provided an upper bound for the freight cost beyond which it becomes unprofitable to place any order with the supplier. Shu et al. [18] studied the impacts of transportation cost on distribution-free newsvendor problems. They formulated the transportation cost as a nonlinear regression function of the shipping quantity. Wangza and Wee [19] included transportation cost into the integrated vendor-buyer inventory model with stochastic demand. Wangza and Wee [20] extended their previous paper to consider freight cost, quantity discount and stochastic demand. Taghizadeh and Venkatachalam [21] considered a multi-item replenishment problem with a piece-wise linear transportation cost under demand uncertainty over a finite planning horizon.

Table 1 classifies existing literature on integrated inventory and transportation models based on multiple dimensions, assisting in positioning our work and highlighting its contributions. The column titled "Transportation cost structure" presents the cost structure adopted by the authors. It indicates whether the authors have explicitly considered the fixed cost per truck and the variable cost that depends on the ordering quantity. It can be observed from Table 1 that most of these models are concentrated on single-echelon environments, based on the assumption of a single-product setting, and exclusively utilized company-owned trucks for transportation.

Compared to the previously reviewed papers, our work introduces three noteworthy contributions. First, we have developed a tighter upper bound for the fixed transportation cost in the context of joint newsvendor and transportation problem when using only owned trucks. For values of the fixed transportation cost below this upper limit, the retailer operational policy is profitable. Our second contribution is the development of a mathematical model and its solution procedure for a common and practical transportation case involving the utilization of either the owned fleet or a mix of owned and leased trucks. Additionally, we have developed an upper bound for the fixed transportation cost of leased trucks. Moreover, the numerical results show that opting for the flexibility and cost advantages of leasing trucks can be an attractive decision for retailers seeking to enhance their transportation operations. Lastly, we have considered the multiple items scenario that enhance the efficient utilization of transportation resources.

## Mathematical models and solution procedures

In this section, we present three integrated newsvendor and transportation problems, each formulated within distinct contexts concerning truck ownership and number of ordered

**Table 2. Listing of the models' parameters.**

| Notation | Definition |
|---|---|
| $c$ | Unit purchasing cost |
| $h$ | Holding cost for each unit of the product unsold by the end of the season |
| $v$ | Unit salvage value |
| $p$ | Unit selling price |
| $\pi$ | Unit shortage cost |
| $m$ | Distance between the supplier and the retailer, measured in miles. |
| $N$ | Number of trucks owned by the retailer |
| $w$ | Truck's loading capacity, measured in units of the product |
| $c_o^f$ | Fixed transportation cost for the owned truck |
| $c_o^v$ | Variable transportation cost for the owned truck |
| $c_l^f$ | Fixed transportation cost for the leased truck |
| $c_l^v$ | Variable transportation cost for the leased truck |
| $GPM_f$ | Fuel consumption of an empty truck, measured in gallons per mile |
| $GPM_e$ | Fuel consumption of a full truck, measured in gallons per mile |
| $c_f$ | Fuel price per gallon |
| $D_w$ | Hourly driver wage |
| $s$ | Average truck speed in miles per hour |
| $M$ | Total number of products for the model involving multiple products |

products. To ensure clarity, we first outline in Tables 2 and 3 the key symbols used across the three models to denote the parameters, and the random and decision variables, respectively.

## Single item newsvendor problem with owned trucks

In this section, we consider the case where the retailer is ordering a single seasonal product, facing a random demand $X$. It is assumed that the retailer and the supplier are operating under the Ex Works (EXW) Incoterm. Under such agreement, the retailer is fully responsible for transporting goods from the supplier's warehouse to its desired destination. This responsibility includes managing all logistics, covering transportation costs, and bearing risks associated

**Table 3. Listing of the models' random and decision variables.**

| Notation | Definition |
|---|---|
| $X$ | Random demand for the product over the selling season with probability density function, $f(x)$, and cumulative distribution function, $F(x)$, having mean $\mu$ and standard deviation $\sigma$ |
| $Q$ | Total ordering quantity to be transported by the trucks fleet |
| $Q_o$ | Quantity transported by owned trucks |
| $Q_l$ | Quantity transported by leased trucks |
| $n_o$ | Number of used owned trucks |
| $n_l$ | Number of trucks to be leased |
| $n$ | Total number of needed trucks to transport the quantity $Q$, $n = n_o + n_l$ |
| $L$ | Auxiliary variable denoting the number of units left in stock at the end of the season = $(Q - X)^+$. Its expected value $E[L] = \int_0^Q (Q - x)dF(x)$ |
| $S$ | Auxiliary variable denoting the number of units sold during the selling season = $\text{Min}(Q, X)$. Its expected value $E[S] = Q - E[L]$ |
| $U$ | Auxiliary variable representing the number of unfulfilled demand units = $(X - Q)^+ = \text{Max}(Q, X) - Q$. Its expected value $E[U] = \mu - Q + E[L]$ |

with the movement of goods. Further information regarding EXW term is available at https://www.investopedia.com/terms/e/exw.asp. Therefore, the retailer exclusively utilizes its homogeneous fleet of $N$ trucks, each capable of carrying a load of $w$ units, to transport the ordered quantity $Q$ from the supplier at the beginning of the season. The incurred transportation costs for operating the fleet of truck consist of fixed costs and variable costs. The fixed costs do not depend on the shipment quantity and include costs such as depreciation, insurance, financing costs, maintenance and repairs, and driver wages. On the other hand, the variable costs are incurred for each unit of the product loaded onto the trucks. These costs mainly depend on the fuel consumption when the truck is fully loaded, fuel consumption when the truck is empty, the number of fully and partially loaded trucks, the quantity loaded on each truck, the distance covered, and the driving speed. Our derivation of the mathematical expressions for the transportation costs are based on the assumption of the linear relationship between fuel consumption and truckload [22]. Given that the retailer is solely relying on its trucks fleet, then $n = n_o$ and $Q = Q_o$. An order of size $Q$ requires $n = \lceil Q/w \rceil$, where $\lceil y \rceil$ is the smallest integer larger than $y$, trucks to be transported with $(n - 1)$ fully loaded trucks and one partially filled truck, carrying $[Q - (n - 1)w]$ units. Therefore, the fuel consumption when transporting $Q$ units, is:

$$m \left\{ (n - 1)GPM_f + GPM_e + \left( \frac{GPM_f - GPM_e}{w} \right)(Q - (n - 1)w) \right\} + m\, n\, GPM_e, \qquad (1)$$

The first term in Eq (1) is the fuel consumption of the $(n - 1)$ fully loaded trucks. The next two terms calculate the fuel consumption of the partially loaded truck transporting $Q - (n - 1)w$ units. The last term is the fuel consumed by the $m$ empty trucks when traveling from the retailer to the supplier.

The fuel consumption cost can be obtained by multiplying the fuel consumption in Eq (1) by $c_f$. After simplification, the fuel consumption cost can be expressed as follows:

$$c_f m \left( \frac{GPM_f - GPM_e}{w} \right) Q + 2\, c_f m\, GPM_e\, n. \qquad (2)$$

Eq (2) consists of two terms, one being dependent on $Q$ and the other that depends on the number of trucks. As in Stellingwerf et al. [23], it is assumed that the driver's wage constitutes the major fixed cost component when compared to other fixed cost elements. Accordingly, the variable and fixed transportation costs can be expressed as follows:

$$C_o^v = c_f m \left( \frac{GPM_f - GPM_e}{w} \right), \qquad (3)$$

$$C_o^f = 2\, c_f m\, GPM_e + D_w \frac{2m}{s}. \qquad (4)$$

The total transportation cost can then be written as

$$c_o^v Q + c_o^f n, \qquad (5)$$

where

$$(n - 1)w < Q \leq nw \qquad (6)$$

Based on Eqs (5) and (6), the total transportation cost function displays a staircase pattern. In the transportation and logistics literature such functional form is commonly known as full truckload, piecewise, or multiple setup cost structure.

Within the context of the joint newsvendor and transportation problems, the goal of the retailer is to determine, prior to the start of the selling season, both the ordering quantity $Q$ and the needed number of trucks that maximize its expected profit, $G(Q, n)$.

$$G(Q, n) = pE[S] + (v - h)E[L] - \pi E[U] - (c + c_o^v)Q - c_o^f n. \tag{7}$$

After substituting the expressions of the various expected values on the right-hand side of Eq (7) and simplifying, the expected profit can be rewritten as follows:

$$G(Q, n) = (p + h - v)\mu - (c + c_o^v + h - v)Q - (p + h + \mu - v)E[U] - c_o^f n. \tag{8}$$

It is clear from Eq (8) that maximizing $G(Q, n)$ is equivalent to minimizing the expected cost function:

$$K(Q, n) = (c + c_o^v + h - v)Q + (p + h + \pi - v)E[U] + c_o^f n, . \tag{9}$$

which can be expressed as

$$K(Q, n) = K^{nvp}(Q) + c_o^f n,$$

where

$$K^{nvp}(Q) = (c + c_o^v + h - v)Q + (p + h + \pi - v)E[U]. \tag{10}$$

is the expected cost of the newsvendor problem under LTL mode of transportation. $K^{nvp}(Q, n)$ is a convex function with an optimal ordering quantity given by:

$$Q^{nvp} = F^{-1}\left(\frac{p + \pi - c - c_o^v}{p + \pi + h - v}\right). . \tag{11}$$

The joint optimization of the newsvendor and transportation problems is stated as follows:

$$Min\ K(Q, n)$$

Subject to:

$$(n - 1)w < Q \leq nw$$

$$n \leq N. \tag{12}$$

$n$ is integer.

The first constraint ensures that the loading capacity of the $n$ trucks is enough for transporting the ordered quantity $Q$. Constraint (12) guarantees that the number of needed trucks to transport $Q$ does not exceed the available number of trucks. Because of the first constraint, both $G(Q, n)$ $K(Q, n)$ are discontinuous functions as illustrated in Fig 1.

Given that $K(Q, n) = K^{nvp}(Q) + c_o^f n$ and $K^{nvp}(Q)$ being a convex function of $Q$, the expected cost function $K(Q, n)$ is strictly decreasing within each of the feasible intervals of $Q$ (i.e., for $Q$ values satisfying $(n - 1)w < Q \leq nw$) when $Q < Q^{nvp}$. The expected cost function also displays a discontinuous jump of $c_o^f$ at each interval boundary. Beyond $Q^{nvp}$, the expected cost function exhibits a strict monotonic increase. Therefore, the optimal ordering quantity, $Q^*$, is either smaller than or equal to $Q^{nvp}$. Moreover, because of the discontinuous jumps at the interval boundaries, if $Q^* \neq Q^{nvp}$, then all the trucks utilized for transporting the optimal ordering quantity are operating at full capacity. We next outline a simple procedure in an algorithmic format to solve the joint optimization of the newsvendor and transportation problems with only owned trucks.

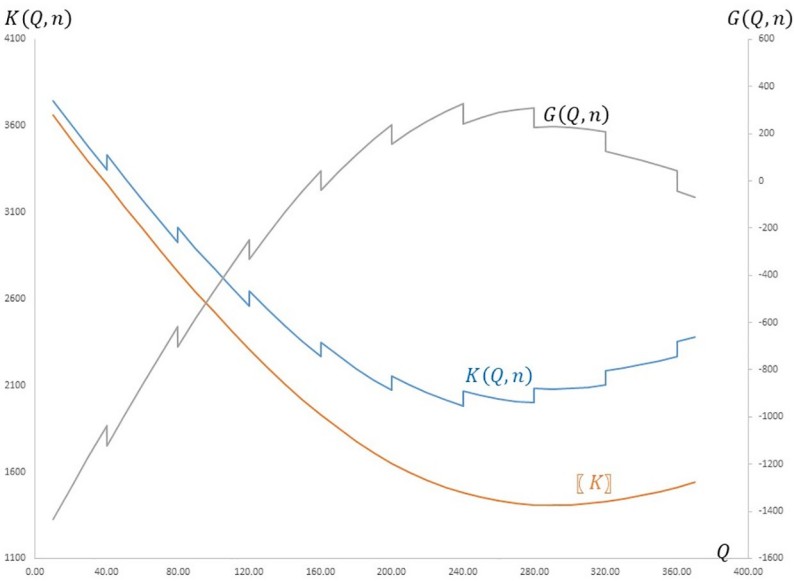

**Fig 1. Graphical representation of the expected profit and cost functions.**

Step 0. Set $K^* = \infty$.
Step 1: Find $Q^{nvp}$ using (11) and $n = \lceil Q^{nvp}/w \rceil$
Step 2: If $n > N$, set $n = N$ and $Q = nw$. Go to step 4.
Step 3. If $n \leq N$, set $Q = Q^{nvp}$ and go to step 4.
Step 4. Compute $K(Q, n)$ using (9)
Step 5. If $K(Q, n) < K^*$, do:
 set $K^* = K(Q, n)$, $Q^* = Q$, and $n^* = n$,
 $n \to n - 1$, $Q = nw$, and go to step 4.
Step 6. If $K(Q, n) \geq K^*$, stop.

In the second step of the solution procedure, all owned trucks are utilized to their maximum loading capacity whenever additional trucks are needed to transport $Q^{nvp}$. On the other hand, the solution procedure starts with $Q^{nvp}$ as an initial solution if the available trucks are sufficient to accommodate this quantity. Then, the algorithm alternates between steps 4 and 5 until it first encounters an increase in the expected cost function.

**Illustrative example 1.**   A retailer is packaging one of its seasonal products in stackable plastic crates. The product is transported using the retailer's fleet of ten small refrigerated trucks, each with a loading capacity of 40 crates. Hereafter, we assume that a single unit of the seasonal product corresponds to one full crate. The seasonal product is facing a random demand, which follows a normal distribution with mean 210 crates and standard deviation of 105 crates. The purchase cost is $3 for each full crate. Any remaining crates in stock at the end of the season costs the retailer $1. The seasonal product is sold at a price of $10 per crate and has no salvage value. However, the retailer incurs a cost $7 for each unit short. Additionally, the fuel consumption for an empty and a full truck are 0.072 and 0.092 gallons per mile, respectively [22]. The travelling distance between the supplier and retailer is100 miles. The cost of one gallon of the diesel fuel used by the truck is $4.00, and the driver's wage is $5 per hour for each owned truck. The average speed of the truck is 60 miles/hour.

Using Eqs (3) and (4), the fixed and variable transportation costs are \$84.43 per truck and \$0.02 per unit, respectively. The steps for finding the optimal ordering quantity and number of trucks are outlined next.

First iteration

Step 1. $Q^{nvp} = 290$ and $n = 8 < 10$.

Step 4. $K(290, 8) = 2080.65$, $n \to 7$

Second iteration

$n = 7$, $Q = 280$, and $K(280, 7) = 1999.22$

$K(280, 7) < K(290, 8)$, $n \to 6$

Third iteration

$n = 6$, $Q = 240$, and $K(240, 6) = 1983.37$

$K(240, 6) < K(280, 7)$, $n \to 5$

Fourth iteration

$n = 5$, $Q = 200$, and $K(200, 5) = 2071.42$

$K(200, 5) > K(240, 6)$, stop.

The algorithm generated the minimum expected cost and its associated optimal ordering quantity and number of trucks in four iterations. The optimal ordering policy is to operate six FTL trucks transporting 240 units with a minimum expected cost of \$1983.37 and a maximum realized expect profit of \$326.63.

Considering the fixed transportation cost as a critical input parameter for the model, we conducted a sensitivity analysis to assess its impacts on the optimal ordering quantity, number of used trucks, and expected profit. The findings of this analysis are detailed in Table 4. As it can be noticed form the table, the newsvendor solution is optimal for smaller values of the fixed ordering cost. However, for values of $c_o^f$ exceeding 3, the retailer will exclusively utilize FTL trucks for transporting the ordering quantity. More precisely, the number of trucks to be operated remains unchanged at 7 FTL trucks when the fixed transportation cost varies between 3 and 68. For values of the fixed transportation costs larger than 68, the number of employed trucks is reduced to 6 FTL trucks. An intuitively apparent result that can also be observed in the same table is that an increase in the fixed transportation cost results in a decrease in the optimal order quantity, the number of trucks required, and the expected profit. This aligns with common logistics practices, as a higher fixed transportation cost incites the retailer to place smaller orders, thereby decreasing the number of trucks required and, as a result, reducing transportation expenses. A final noteworthy observation is that when the fixed

**Table 4. Impacts of $c_o^f$ on optimal ordering quantity, number of used trucks, and expected profit.**

| $c_o^f$ | $n^*$ | $Q^*$ | $G(Q^*, n^*)$ |
|---|---|---|---|
| 1 | 8 | 290 | 893.4 |
| 2 | 8 | 290 | 885.4 |
| 3 | 7 | 280 | 877.8 |
| 20 | 7 | 280 | 758.8 |
| 50 | 7 | 280 | 584.8 |
| 68 | 7 | 280 | 422.8 |
| 69 | 6 | 240 | 416.6 |
| 100 | 6 | 240 | 230.6 |
| 120 | 6 | 240 | 110.6 |
| 138 | 6 | 240 | 2.6 |
| 139 | 6 | 240 | -3.4 |
| 200 | 6 | 240 | -341.4 |

transportation cost $c_o^f$ exceeds 138, relying solely on owned trucks is no longer economically viable for the retailer.

Zhang et al. [6] showed that when the fixed transportation cost exceeds $[K^{nvp}(0) - K^{nvp}(\min(w, Q^{nvp}))]$, then it is optimal not to place any orders. However, this condition does not ensure the retailer's profitability. It is easy to verify that for $c_o^f$ values smaller than $K^{nvp}(0) - K^{nvp}(\min(w, Q^{nvp}))$, the optimal policy is also not profitable. Indeed, taking the same illustrative example, $K^{nvp}(0) - K^{nvp}(40) = 517.2$, yet it remains unprofitable to operate the owned fleet of trucks for $c_o^f$ values within the range of $(139, 517.2)$. Determining the maximum allowable value for the fixed transportation cost, which would result in positive profit is equivalent to finding the maximum expected profit per truck. Therefore, to guarantee the economic viability of the ordering policy, the fixed transportation cost must be smaller than the maximum expected profit per truck. This optimization problem can be expressed mathematically as follows:

$$Max \left[ (p + h - v)\mu - \left(c + c_o^v + h - v\right)Q - (p + h + \pi - v)E[U] \right] / n$$

Subject to:

$$(n - 1)w < Q \le nw$$

$$n \le N$$

$n$ is integer.

The same solution procedure outlined earlier can be applied to solve this optimization problem. For the illustrative example, the maximum expected profit per truck amounts to $138.4385. Consequently, the retailer should not operate its owned feet of trucks for any fixed transportation cost exceeding $138.4385.

## Single item newsvendor problem with owned and leased trucks

The retailer is now considering the transportation option that involves operating a mix of owned and leased trucks to handle larger ordering quantities. Opting for leasing trucks provides the retailer with the advantage of accessing additional trucks as required, without the need to invest in acquiring new ones. This option is particularly appealing when facing fluctuating transportation needs because of the random demand, as leasing contracts can be tailored to fit specific durations of the selling season. Hereafter, we assume that transportation costs for leased trucks exceed those of owned trucks. This assumption is realistic to prioritize the operation of owned trucks initially and consider the leasing option only when additional loading capacity is required. Moreover, without loss of generality, we also assume that owned and leased trucks are of the same type having identical loading capacities. Lastly, the retailer has no restriction on the number of trucks that can be leased from the market.

The expected cost for operating both owned and leased trucks can be written as:

$$K(Q_o, Q_l, n_o, n_l) = (c + h - v)Q + (p + h + \pi - v)E[U] - c_o^v Q_o - c_l^v Q_l - c_o^f n_o - c_l^f n_l \quad (13)$$

The joint newsvendor and transportation problem with owned and leased trucks is then mathematically expressed as follows:

$$Min\, K(Q_o, Q_l, n_o, n_l)$$

Subject to:

$$Q_o + Q_l = Q \tag{14}$$

$$(n_o - 1)w < Q_o \leq n_o w \tag{15}$$

$$(n_l - 1)w < Q_l \leq n_l w \tag{16}$$

$$n_o \leq N \tag{17}$$

$n_o$ and $n_l$ are integers.

Constraint (14) states that the ordering quantity is the sum of the quantities transported in the owned and leased trucks. Constraints (15) and (16) are similar to constraint (6). Constraints (17) ensures that the number of utilized owned trucks is smaller than the number of available trucks. The following lemma provide a structural property of the optimal solution.

**Lemma 1**: In an optimal solution to the integrated newsvendor and transportation problem with owned and leased trucks, leased trucks are only utilized when the loading capacity of the owned trucks is not sufficient to transport the ordering quantity.

**Proof**: The proof is straightforward based on the transportation costs assumption for owned and leased trucks.

Using Lemma 1, $Q_l = 0$ and $Q_l = Q_o$ when $n_o \leq N$. Moreover, in this case, the expected cost is determined as follows:

$$K(Q, 0, \ n_o, 0) = (c + h - v)Q + (p + h + \pi - v)E[U] - c_o^v Q - c_o^f n_o. \tag{18}$$

The same solution procedure developed in the previous section can be used to find the optimal ordering quantity and number of owned trucks to be used. However, when $n_o > N$, the owned trucks can only accommodate $Nw$ units and the remaining quantity, $Q_l = Q - N_w$, are transported by the leased trucks. The expected cost is then given by:

$$K(nw, Q_l, \ N, n_l) = (c + h - v)Q + (p + h + \pi - v)E[U] + c_o^v Nw + c_l^v Q_l + c_o^f N + \ c_l^f n_l \tag{19}$$

In such case, the optimal unconstrained newsvendor quantity is:

$$Q_l^{nvp} = F^{-1}\left(\frac{p + \pi - c - c_l^v}{p + \pi + h - v}\right). \tag{20}$$

The ordering quantity minimizing the expected cost in (19) should satisfy:

$$Nw + (n_l - 1)w < Q(N, \ n_l) \leq Nw + n_l w. \tag{21}$$

We propose the following algorithm to solve the integrated newsvendor and transportation optimization problem involving both owned and leased trucks. It consists of two stages. During the first stage, it solves the optimization problem with only owned trucks. In case the number of trucks obtained by the first stage is not feasible ($n_o > N$), the second phase of the algorithm, utilizing $N$ owned truck, is activated to find the number of leased trucks $n_l$.

***First Stage***:

Step 0. Set $K^* = \infty$.

Step 1: Find $Q^{nvp}$ using (11) and $n = \lceil Q^{nvp}/w \rceil$

Step 2: If $n > N$, set $n = N$ and $Q = nw$. Go to step 4.

Step 3. If $n \leq N$, $Q = Q^{nvp}$. Go to step 4.

Step 4. Compute $K(Q, n)$ using (9)

Step 5. If $K(Q, n) < K^*$, do:

   set $K^* = K(Q, n)$, $Q^* = Q$, and $n^* = n$,

   $n \to n - 1$, $Q = nw$, and go to step 4.

Step 6. If $K(Q, n) \geq K^*$, do:

   $n_o^* = n^*$, $Q_o^* = Q^*$

   Go to the second stage.

***Second Stage***:

Step 7: If $n_o^* < N$, go to step 13

Step 8: If $n_o^* = N$, do:

   Set $K^* = \infty$.

   $Q_o^* = Nw$

   Find $Q_l^{nvp}$ using (20)

   Set $Q_l = Q_l^{nvp} - Nw$, $n_l = \left\lceil \frac{Q_l}{w} \right\rceil$

   Go to step 10

Step 9: If $n_l < 0$, go to step 13

Step 10: Compute $K(Nw, Q_l, N, n_l)$ using (19)

Step 11: $K(Nw, Q_l, N, n_l) < K^*$, do:

   Set $K^* = K(Nw, Q_l, N, n_l)$, $Q_l^* = Q_l$, and $n_l^* = n_l$,

   $n_l \to n_l - 1$, $Q_l = n_l w$, and go to step 10.

Step 12. $K(Nw, Q_l, N, n_l) \geq K^*$, go to step 13

Step 13: Stop

The second stage of the algorithm is executed only when the optimal number of owned trucks obtained in the first stage is equal to the maximum available number of trucks. In this case, all owned trucks are loaded to their full capacity ($Q_o^* = Nw$ and $n_o^* = N$). During this second stage, the algorithm decreases iteratively the number of leased trucks by one until either the expected cost increases for the first time or the number of leased trucks reaches zero.

**Illustrative example 2.** Suppose that the retailer now has only 4 trucks available from its owned fleet of trucks to transport the ordering quantity from the supplier. The retailer is also considering leasing trucks with same loading capacity, but with fixed and variable transportation costs of \$95 per truck and \$0.05 per unit, respectively. All other remaining data for the problem are kept the same as in the previous illustrative example. The results of the different steps of the algorithm are outlined below.

**First Stage**:

Step 1. $Q^{nvp} = 290$ and $n = 8 > 4$.

Step 2. $n = 4$ and $Q = 160$,

Step 4. $K(160, 4) = 2267.11$

Step 5. $n \to 3$, $Q = 120$

Step 4. $K(120, 3) = 2559.59$

Step 6. $n_o^* = 4$, $Q_o^* = 160$, $K(160, 4) = 2267.11$, and $G(160, 4) = 42.89$

**Second Stage**:

Step 8: $Q_o^* = 160$, $Q_l^{nvp} = 290$

   Set $Q_l = 130$, $n_l = 4$.

Step 10: $K(160, 130, 4, 4) = 2128.53$

Step 11: $K(160, 130, 4, 4) = 2128.53 < K^* = 2267.11$

   Set $n_l = 3$, $Q_l = 120$

Step 10: $K(160, 120, 4, 3) = 2035.81$

Step 11: $K(160, 120, 4, 3) = 2035.81 < K(160, 130, 4, 4)$

   Set $n_l = 2$, $Q_l = 80$

**Table 5. Impacts of $c_l^f$ on optimal ordering quantity, number of leased trucks, and expected profit.**

| $c_l^f$ | $n_l^*$ | $Q_l^*$ | $G(160, Q_l^*, 4, n_l^*)$ |
|---|---|---|---|
| 10 | 3 | 120 | 529.18 |
| 40 | 3 | 120 | 439.18 |
| 60 | 3 | 120 | 379.18 |
| 70 | 2 | 80 | 352.23 |
| 100 | 2 | 80 | 292.23 |
| 170 | 2 | 80 | 152.23 |
| 180 | 1 | 40 | 141.38 |
| 250 | 1 | 40 | 71.38 |
| 275 | 1 | 40 | 46.38 |
| 278 | 1 | 40 | 43.38 |
| 279 | 0 | 0 | 42.89 |

Step 10: $K(160, 80, 4, 2) = 2007.77$
Step 11: $K(160, 80, 4, 2) = 2007.77 < K(160, 120, 4, 3)$
     Set $n_l = 1$, $Q_l = 40$
Step 10: $K(160, 40, 4, 1) = 2083.62$
Step 12: $K(160, 40, 4, 1) = 2083.62 > K(160, 80, 4, 2)$
     Stop

The optimal solution is to place an order for 240 units, out of which 160 units are transported by 4 owned trucks, while the remaining 80 units are shipped using 2 leased trucks, resulting in an expected cost of $2007.77 and an expected profit of $302.23. By deciding to lease trucks, the retailer can save $259.23.

A sensitivity analysis is carried out to evaluate the impacts of the fixed transportation cost on the joint inventory and transportation policy. Table 5 reveals that the fixed transportation cost associated with leasing trucks has a diminishing effect on the optimal ordering quantity, number of leased trucks, and expected profit. Moreover, leasing trucks does not yield any additional profit over using exclusively owned trucks when $c_l^f$ values exceed $278. This breakeven point for the fixed transportation cost of leased trucks is equal to the maximum additional profit per truck that can be achieved through leasing trucks. The maximum added profit per one leased truck can be obtained by solving the following optimization problem:

$$Max \left[ (p + h - v)\mu - (c + c_o^v + h - v)Q - (p + h + \pi - v)E[U] \right.$$
$$\left. - c_o^v Nw - c_o^f N - c_l^v (Q - Nw) - G(Nw, 0, N, 0) \right] / n_l$$

Subject to:

$$Nw + (n_l - 1)w < Q \le Nw + n_l w$$

$n_l$ is integer.

## Multi-item newsvendor problem with owned trucks

In this section, the retailer is managing the inbound transportation of multiple items and their ordering quantities from the same supplier. Its goal is to determine the optimal quantity of each item to be ordered prior to beginning of the season and the required number of trucks to transport all the items that maximize its total expected profit. We assume that all products are packed in the same crates/boxes of the same size to efficiently utilize the transportation and

storage capacities. This packing practice is particularly prevalent for various vegetable and fruit products with similar shape and weights, like tomatoes, onions, apples, and oranges. These products are commonly packed in crates with identical dimensions to ensure efficient utilization of truck capacity. Furthermore, it is assumed that the sales of one transported product does not predict or affect the sales of the other. In other words, it is assumed that there is no significant correlation between the sales of the transported products.

The optimization problem for the multi-item case can be mathematically formulated as follow:

$$Max \sum_{j=1}^{M} \left[ \left[ \left( p_j + h_j - v_j \right) \mu_j \right] \right] - \sum_{j=1}^{M} K_j \left( Q_j \right) - c_o^v Q - n c_o^f \qquad (22)$$

Subject to:

$$\sum_{j=1}^{M} Q_j = Q \qquad (23)$$

$$(n-1)w < Q \leq nw$$

$$n \leq N$$

$n$ is integer,
where:

$$K_j \left( Q_j \right) = \left( c_j + h_j - v_j \right) Q_j + \left( p_j + h_j + \pi_j - v_j \right) E \left[ U_j \right].$$

In the following, we consider minimizing the expected cost function:

$$K \left( Q_j, n \right) = \sum_{j=1}^{M} \left[ \left( c_j + c_o^v + h_j - v_j \right) Q_j + \left( p_j + h_j + \pi_j - v_j \right) E \left[ U_j \right] \right] + n c_o^f$$

subject to the same constraints.

For a given number of trucks, $n$, the objective and constraints of the cost optimization problem are convex. Therefore, the Karush-Kuhn-Tucker (KKT) conditions are necessary and sufficient conditions for the optimal solution. These conditions are as follows:

$$c_o^v + c_j - p_j - \pi_j + \left( p_j + h_j + \pi_j - v_j \right) F \left( Q_j \right) + \lambda - \gamma = 0 \text{ for } j = 1, \ldots, M. \qquad (24)$$

$$\lambda \left( \sum_{j=1}^{M} Q_j - nw \right) = 0 \qquad (25)$$

$$\gamma \left( (n-1)w - \sum_{j=1}^{M} Q_j \right) = 0 \qquad (26)$$

$$\lambda \geq 0 \text{ and } \gamma \geq 0$$

where $\lambda$ and $\gamma$ are the Lagrangian multipliers of the constraints $Q \leq nw$ and $(n-1)w < Q$, respectively.

Assuming that not ordering any of the items ($Q = 0$) is a sub-optimal solution, then it is easy to show that in an optimal solution, $\gamma$ must equal to 0 for any value of $n$. In this case, KKT conditions reduce to Eq (25) and the following equation:

$$Q_j(n, \lambda) = F^{-1}\left[\left(p_j + \pi_j - c_j - c_o^v - \lambda\right)/\left(p_j + h_j + \pi_j - v_j\right)\right] \tag{27}$$

Consider the case $\lambda = 0$. From (27), we have

$$Q_j(n, 0) = F^{-1}\left[\frac{p_j + \pi_j - c_j - c_o^v}{p_j + h_j + \pi_j - v_j}\right] = Q_j^{nvp}. \tag{28}$$

In case $\sum_{j=1}^{M} Q_j(n, 0) \leq nw$, then $\{Q_j(n, 0), j = 1, \ldots, M\}$ is optimal for this number of trucks, $n$.

Suppose now $\sum_{j=1}^{M} Q_j(n, 0) - nw > 0$. Then, we need find a positive $\lambda$ to satisfy the KKT condition (25). This positive $\lambda$ is the solution to the following equation:

$$\sum_{j=1}^{M} Q_j(n, \lambda) - nw = 0. \tag{29}$$

Let $Z(\lambda)$ be the left-hand side of Eq (29). Then,

$$\frac{dZ(\lambda)}{d\lambda} = \sum_{j=1}^{M} \frac{dQ_j(n, \lambda)}{d\lambda}.$$

Using Eq (27), we get

$$\frac{dZ(\lambda)}{d\lambda} = -\frac{1}{\left(p_j + h_j + \pi_j - v_j\right)f\left(Q_j(n, \lambda)\right)} < 0.$$

Therefore, $Z(\lambda)$ is a strictly decreasing function of $\lambda$.

Given that $Z(0) > 0$, we need to determine a positive value of $\lambda$ that leads to a negative $Z(\lambda)$ in order to solve Eq (29). Subsequently, we use the bi-section method to find $\lambda(n)$ such that $Z(\lambda(n)) = 0$. It is worth noting that during the search for $\lambda(n)$, if $\lambda \geq p_j + \pi_j - c_j - c_o^v$ for item $j$, we set $Q_j(n, \lambda) = 0$ because $0 \leq F(Q_j(n, \lambda))$ in Eq (27). In the following, we present the different steps of the algorithm to solve the multi-item newsvendor problem with owned trucks.

Step 0. Set $K^* = \infty$.

Step 1: Find $Q_j^{nvp}$ using (28) for $j = 1, 2, \ldots, M$, $Q^{nvp} = \sum_{j=1}^{M} Q_j^{nvp}$, and $n = \lceil Q^{nvp}/w \rceil$

Step 2: If $n > N$, set $n = N$. Go to step 4.

Step 3. If $n \leq N$ set $Q = Q^{nvp}$ and go to step 5.

Step 4. Solve $\sum_{j=1}^{M} Q_j(n, \lambda) - nw = 0$ using the bisection method to get $\lambda(n)$.

Set $Q_j = Q_j(n, \lambda(n))$ and $Q = \sum_{j=1}^{M} Q_j$. Go to step 5.

Step 5. Compute

$$K\left(Q_j, n\right) = \sum_{j=1}^{M} \left[\left(c_j + c_o^v + h_j - v_j\right)Q_j + \left(p_j + h_j + \pi_j - v_j\right)E\left[U_j\right]\right] + nc_o^f$$

Step 5. If $K(Q_j, n) < K^*$, do:

set $K^* = K(Q, n)$, $Q^* = Q$, and $n^* = n$,

$n \rightarrow n - 1$, $Q = nw$, and go to step 4.

Step 6. If $K(Q, n) \geq K^*$, stop.

**Table 6. Items data.**

| Item | 1 | 2 | 3 | 4 |
|------|---|---|---|---|
| $p$ | 10 | 12 | 8 | 5 |
| $c$ | 3 | 4 | 2 | 1 |
| $h$ | 1 | 1 | 1 | 1 |
| $\pi$ | 7 | 8 | 4 | 3 |
| $\nu$ | 0 | 0 | 0 | 0 |
| $\mu$ | 210 | 180 | 120 | 70 |
| $\sigma$ | 80 | 70 | 50 | 30 |

**Table 7. Independent optimal solutions.**

| Item | Number of trucks | Ordering quantity | Cost |
|------|------------------|-------------------|------|
| 1 | 3 | 240 | 1561.2 |
| 2 | 3 | 230 | 1611.59 |
| 3 | 2 | 157 | 728.87 |
| 4 | 1 | 80 | 314.24 |

The steps of the algorithm are similar to the ones employed for the single item problem, with the addition of the bisection method used in step 4 to find the ordering quantities that make their sum equal to the available loading capacity of $n$ trucks.

**Illustrative example 3.** To efficiently utilize its transportation resources, the retailer has decided to consolidate the ordering of four products from a single supplier. The fixed and variable costs associated with each the ten available trucks for transporting these four items from the supplier are $84 per truck and $0.02 per unit, respectively. The truck's loading capacity is 80 units. The relevant data for the four items is displayed in Table 6.

In step 1 of the algorithm, the newsvendor quantities were (271, 230, 156, 157, 93). The total quantity to be transported is 751, requiring the deployment of 10 trucks. This shipping plan results in an overall cost of $4256.94. After reducing the number of trucks to 9 trucks and solving the equation $\sum_{j=1}^{M} Q_j(n, \lambda) - nw = 0$, the ordering schedule is (262, 223, 149, 86), resulting in an overall cost of $4182.3. As the overall cost is decreased, we reduce the number of trucks once again to 8 trucks. Upon completing step 4 of the algorithm, we obtain the ordering quantities (237, 205, 129, 69), incurring an overall cost of $4224.90. At this point, the algorithm terminates because of the increase in the overall cost. Consequently, the optimal solution is to utilize 9 FTL trucks at an overall cost of $4182.3.

When the four items are ordered individually, their respective optimal ordering policies are shown in Table 7. As can be observed from the same table, the joint replenishment of the four orders led to a cost savings of $33.6 and better utilization of the trucks' capacities. Upon consolidation of the orders, all 9 trucks were fully utilized. In contrast, when the items were ordered independently, 7 FTL and 2 LTL trucks were employed.

## Conclusion

This paper presents three main contributions along with their managerial implications. We first develop a mathematical model and propose a simple iterative solution procedure when the retailer depends solely on its owned fleet of trucks. For such situation, it is crucial to monitor the fixed transportation costs to ensure they do not exceed the breakeven point, beyond

which the retailer would fail to realize any profit. The retailer should carefully monitor the various components of fixed costs—including insurance, financing costs, maintenance, and repairs—to identify and implement cost reduction strategies. We provide a lower breakeven point for the fixed transportation cost than previously proposed in the literature. Additionally, we investigate the flexible transportation case, wherein the retailer has the option to lease extra trucks from the external market as needed. Such flexibility allows the retailer to handle larger ordering quantities and leading to savings in costs. Finally, we address the problem of efficiently transporting multiple seasonal products by operating exclusively the retailer's owned trucks. We propose an iterative algorithm based on the Lagrangian method to optimally solve this complex optimization problem. The joint replenishment of multiple items enables the retailer to realize cost savings and enhance the utilization of its truck fleet.

A potential extension of the problems addressed in this paper entails an examination of the environmental impacts of the storage and inbound transportation activities when using retailer-owned and leased trucks. Even though leasing newer and more fuel-efficient trucks could lead to higher fixed and variable transportation costs, it has the potential to reduce carbon emissions, ultimately cutting down the overall carbon footprint of the supply chain. Another possible extension is the exploration of outsourcing either a part or the whole inbound transportation process, rather than leasing trucks. This outsourcing could be achieved through collaborations with experienced third-party logistics providers. In addition, examining the impact of transportation costs on the effectiveness of collaboration initiatives, like vendor managed inventory and continuous replenishment [24], is a valuable avenue for future research. Considering the purchase price as a decision variable instead of an externally determined parameter, in the context of general price-dependent demand, presents an interesting extension. This research extension enables the adjustment of prices to compensate for high fixed transportation costs, thereby ensuring profitability. Finally, investigating the impact of supplier discounts on transportation costs for large orders merits further investigation in the future.

## Supporting information

**S1 Data. Data to reproduce the curves in Fig 1.**
(XLSX)

## Acknowledgments

The author is grateful to the esteemed academic editor and reviewers for their valuable feedback, which significantly enhanced the content, quality, and clarity of the paper. This paper represents the opinions of the author and does not mean to represent the position or opinions of the American University of Sharjah.

## Author Contributions

**Conceptualization:** Moncer Hariga.

**Formal analysis:** Moncer Hariga.

**Methodology:** Moncer Hariga.

**Writing – original draft:** Moncer Hariga.

**Writing – review & editing:** Moncer Hariga.

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
