## [Decision Letter · Decision Letter 0]

12 Mar 2024

PONE-D-24-00923Incorporating transportation costs into the single and multiple items newsvendor problemsPLOS ONE

Dear Dr. Hariga,

Thank you for submitting your manuscript to PLOS ONE. After careful consideration, we feel that it has merit but does not fully meet PLOS ONE’s publication criteria as it currently stands. Therefore, we invite you to submit a revised version of the manuscript that addresses the points raised during the review process.

We look forward to receiving your revised manuscript.

Kind regards,

Mohammad Heydari

Academic Editor

PLOS ONE

Journal Requirements:

6. Please include a copy of Table 2 which you refer to in your text on page 15.

**Additional Editor Comments:**

Dear Authors,

I hope this letter finds you well. I am writing to inform you of the editorial decision regarding your manuscript titled "Incorporating Transportation Costs into the Single and Multiple Items Newsvendor Problems," submitted to PLOS ONE under Manuscript ID PONE-D-24-00923.

Firstly, I would like to express our sincere appreciation for your contribution to our journal and for your patience throughout the review process. Your work addresses an important topic in our field, and we value the effort you have put into your research.

After careful consideration of the manuscript and the comprehensive feedback provided by our reviewers, it has been decided that your submission requires a major revision before it can be accepted for publication. The reviewers have highlighted specific areas that need attention, and we believe that addressing these concerns will significantly enhance the quality and clarity of your work.

I have attached the reviewers' comments along with the revised version of your manuscript. Please carefully review the comments and make the necessary revisions to address the concerns raised. Ensure that you provide a detailed response to each comment in the cover letter accompanying the revised manuscript.

The revised manuscript, along with the response to reviewers, should be submitted through the online submission system in a month from this date.

We appreciate your commitment to enhancing the rigor and impact of your work, and we look forward to receiving your revised manuscript. Should you have any questions or need further clarification on the reviewers' comments, please do not hesitate to contact us.

Thank you for choosing PLOS ONE for the dissemination of your research, and we appreciate your cooperation in this important stage of the publication process.

Best regards,

Mohammad Heydari

Reviewers' comments:

Reviewer's Responses to Questions

**Comments to the Author**

1. Is the manuscript technically sound, and do the data support the conclusions?

Reviewer #1: No

Reviewer #2: Partly

Reviewer #3: Partly

2. Has the statistical analysis been performed appropriately and rigorously? 

Reviewer #1: N/A

Reviewer #2: Yes

Reviewer #3: No

3. Have the authors made all data underlying the findings in their manuscript fully available?

Reviewer #1: No

Reviewer #2: Yes

Reviewer #3: Yes

4. Is the manuscript presented in an intelligible fashion and written in standard English?

Reviewer #1: Yes

Reviewer #2: Yes

Reviewer #3: Yes

5. Review Comments to the Author

Reviewer #1: In this manuscript, the author presented a number of newsboy models associated with the transprtation cost, which is not novel since they only explicitly express the purchasing cost, as said by the author. With the ideas and findings in this manuscript, my main concerns are stated as follows.

(1) The decision-making subject of the newsboy problem (retailer) is usually to determine the optimal replenishment quantity and optimal price. Transportation strategy is the responsibility of suppliers. In practice, retailers place more orders, which may reduce the transportation costs of suppliers, then there are different discount of purchasing price. I think that the built models in this manuscript are the similar ones with discounts.

If not, the author MUST state the reasons of doing so in this manuscript.

(2) Is it reasonable for fuel cost to be a linear function of loading capacity? What are the other papers besides the author's own paper? In addtion, if transportation decisions really need to be made by retailers, not only should the demand be uncertain, but also the uncertainty of unit transportation costs should be considered (traffic conditions are complex);

(3) The decision variable in the cost function (objective function) is only Q, and n depends on Q. They are not two decision variables and are expressed incorrectly;

(4) Is the optimal replenishment quantity of equation (11) obtained by taking the derivative method correct due to the presence of integer intermediate variables in the built model?

(5) When considering using self owned or rented cars, their respective cost structures are different, and it should be possible to optimize the optimal proportion of each transportation method? rather than the results in Lemma 1.

(6) If the multi-product newsboy problem does not consider the correlation in product sales and the impact of their different sizes on the empty space of the carriage when transporting and loading them, then the constructed multi product newsboy model has little significance (directly adding up), i.e. no considering joint distribution of random demands.

Reviewer #2: Major Comments

1.This study proposed three models in different situations. However, the authors did not explain the relationship between these models or compare the results of these models. It is also suggested to explain why three different circumstances are considered.

2.In Section 3, in order to facilitate the explanation of the models, I suggest that the authors should organize and display the symbols used in the models in a table and distinguish which ones are parameters and which are decision variables.

3.In Table 3, why are the values of v for all items 0?

4.In numerical example analysis, is it possible to perform sensitivity analysis of parameters?

5.In Section 6, the authors should highlight the academic or theoretical contribution based on the results of the analysis and compare with the results of previous relevant studies to increase the completeness of this study.

Minor Comments

1.The authors should explain how Figure 1 was drawn. That is, what are the parameter values of these functions?

2.Please confirm the number of the form again.

3.Page 7, line↑7，”(, )” should be “()”.

4.Page 15, lines 1-6, the solution process for stage 2 is confusing and the authors are advised to state it specifically.

Reviewer #3: 1. Include a brief mention of the results in the abstract.

2. Add a table detailing author contributions.

3. Number the provided examples.

4. Ensure that order quantity in Table 4 consists of whole numbers.

5. Expand the conclusion section with detailed explanations and include managerial implications.

6. Include sensitivity analysis of the models.

6. PLOS authors have the option to publish the peer review history of their article (what does this mean?). If published, this will include your full peer review and any attached files.

Reviewer #1: **Yes: **Zhong Wan

Reviewer #2: No

Reviewer #3: **Yes: **Mandeep Mittal

---

## [Author Response · Author response to Decision Letter 0]

24 Apr 2024

Journal Requirements

 PLOS ONE's style requirements

We have revised the manuscript to the best of our ability to meet PLOS ONE's style requirements.

 Please include a copy of Table 2 which you refer to in your text on page 15.

There was a typo in the Table’s caption, which has been corrected. 

Reviewer #1: 

In this manuscript, the author presented a number of newsboy models associated with the transprtation cost, which is not novel since they only explicitly express the purchasing cost, as said by the author. 

Response: The author did not only explicitly express the purchasing costs. It was mentioned in the paper "However, in this paper, we explicitly incorporate both per-unit and per-truck transportation costs into the single and multiple items newsvendor problems". The author meant that unlike most of newsvendor problems encountered in literature which tend to include per-unit transportation costs implicitly within the purchasing costs, this paper explicitly incorporates both per-unit and per-truck transportation costs into its models. To clarify, we have revised the statement and invite the reviewer to refer to the second paragraph on page 2 of the updated version.

With the ideas and findings in this manuscript, my main concerns are stated as follows.

(1) The decision-making subject of the newsboy problem (retailer) is usually to determine the optimal replenishment quantity and optimal price. Transportation strategy is the responsibility of suppliers. In practice, retailers place more orders, which may reduce the transportation costs of suppliers, then there are different discount of purchasing price. I think that the built models in this manuscript are the similar ones with discounts.

If not, the author MUST state the reasons of doing so in this manuscript.

Response: We appreciate the reviewer's comments highlighting key areas requiring further clarification. We concur that in certain situations, decision-makers may have to jointly determine the optimal inventory replenishment quantity and pricing strategy. This adjustment is particularly needed in instances of high fixed transportation costs to ensure the retailer’s profitability. However, incorporating price as a decision variable extends beyond the present paper's scope, as it would significantly complicate the model by introducing price-dependent stochastic demand. We have suggested considering price as a decision variable for future research. Please see the conclusion section for further details.

The transportation strategy can be the responsibility of the supplier or the retailer depending on the terms of the transportation contract. Under the Ex Works (EXW) agreement, as defined by Incoterms, the retailer is fully responsible for transporting goods from the supplier's warehouse to its desired destination. This responsibility includes managing all logistics, covering transportation costs, and bearing risks associated with the movement of goods. The EXW term places minimum commitments on the supplier, with the retailer taking on most of the transportation costs and risks. For further reading, please visit https://www.investopedia.com/terms/e/exw.asp. Moreover, the trend of retailers relying on their own truck fleets for transportation from suppliers to their warehouses is on the rise. This strategy enhances control over the supply chain, cuts costs, and ensures more reliable delivery schedules. Indeed, it is common for retailers to place larger orders, potentially reducing the transportation costs for suppliers, who may then offer discounts on the purchase price to the retailer. However, placing more orders can lead to an increased number of trucks required and, consequently, higher transportation costs, which might offset any discounts received on purchasing costs. This is also an interesting problem worth considering in the future. Please see the conclusion section

In conclusion, the models developed in this paper are not related to quantity-discount problems. 

(2) Is it reasonable for fuel cost to be a linear function of loading capacity? What are the other papers besides the author's own paper? In addtion, if transportation decisions really need to be made by retailers, not only should the demand be uncertain, but also the uncertainty of unit transportation costs should be considered (traffic conditions are complex);

Response: Several papers in the literature made a reasonable assumption about the linear relationship between fuel consumption and load weight. As highlighted in the paper, Reference [22] also adopted this assumption. Additionally, to mention just a few, here are few papers that made a similar assumption.

 P. Gajendran, N.N. Clark. Effect of truck operating weight on heavy-duty diesel emissions. Environ. Sci. Technol., 37 (18) (2003), pp. 4309-4317

 S. Ubeda, F.J. Arcelus, J. Faulin. Green logistics at Eroski: a case study. Int. J. Prod. Econ., 131 (1) (2011), pp. 44-51

 Ali Bozorgi, Jennifer Pazour, Dima Nazzal. A new inventory model for cold items that considers costs and emissions, International Journal of Production Economics, vol. 155, 2014, Pages 114-125,

 In this paper, we have listed the various components of fixed and variable unit transportation costs, which can be accurately estimated in practice. Please see the first paragraph on page 8 for more information. To avoid traffic congestion, it is a common practice for trucks to depart from the supplier’s warehouse in the early hours (4 am) of the morning and are assigned early delivery time windows at the retailer’s end (6am – 7am). This approach is effective in reducing the randomness caused by traffic congestion.

(3) The decision variable in the cost function (objective function) is only Q, and n depends on Q. They are not two decision variables and are expressed incorrectly;

Response: Q serves as the primary decision variable, while n functions as an auxiliary variable that has to satisfy the inequalities in (6). In particular, n is dependent on Q based on the relation n=⌈Q/w⌉, where ⌈x⌉ is the smallest integer larger than x. However, this relationship cannot be directly incorporated into the optimization problem; instead, it is accounted for by incorporating the inequalities specified in (6).

(4) Is the optimal replenishment quantity of equation (11) obtained by taking the derivative method correct due to the presence of integer intermediate variables in the built model?

Response: Please note the optimal replenishment quantity in equation (11), Q^nvp, is applicable to the classical newsvendor problem under LTL mode of transportation and without considering the fixed transportation cost. In other words, there is only one decision variable, Q and n is not as decision variable in the newsvendor problem. 

(5) When considering using self owned or rented cars, their respective cost structures are different, and it should be possible to optimize the optimal proportion of each transportation method? rather than the results in Lemma 1.

Response: Please note that the ordering quantity is a decision variable is not predetermined so that one can optimize the optimal proportion of each transportation method. Indeed, if Q is known and is less than Nw, then the use of owned trucks exclusively is needed. However, when it exceeds Nw, then both of types of trucks have to be used. Therefore, one can determine which policy to use without knowing Q. 

(6) If the multi-product newsboy problem does not consider the correlation in product sales and the impact of their different sizes on the empty space of the carriage when transporting and loading them, then the constructed multi product newsboy model has little significance (directly adding up), i.e. no considering joint distribution of random demands.

Response: As mentioned in the paper, it is assumed that all products are packed in identical pallets, crates, or boxes of uniform size, to efficiently utilize the transportation and storage capacities. This packing practice is especially common among various vegetable and fruit products with similar shape and weights, like tomatoes, onions, apples, and oranges. These products are commonly packed in crates with identical dimensions to ensure efficient utilization of truck capacity. It is also assumed that the sales of one transported product does not predict or affect the sales of the other. In other words, it is assumed that there is no significant correlation between the sales of the transported products. Indeed, specific vegetables and fruits (e.g., onions and oranges) are sold independently of one another as they are influenced by different factors like seasonality and consumer preferences, rather than exhibiting a significant correlation in sales.

Many thanks again for the constructive comments and suggestions

Reviewer #2: Major comments

1.This study proposed three models in different situations. However, the authors did not explain the relationship between these models or compare the results of these models. It is also suggested to explain why three different circumstances are considered.

Response: In the abstract and the last paragraph of the introduction section, we explain the progression and interconnection among three distinct yet related models within the scope of an integrated inventory and transportation problem. Initially, we formulate a foundational model where a retailer is transporting a single product, utilizing solely its own fleet of trucks. In the second model, we expand the retailer’s logistics resources by considering the option to lease trucks. This modification allows for a more flexible transportation strategy, accommodating scenarios where the retailer's own fleet may be insufficient due to capacity constraints. The third model further extends the operational scope by relaxing the single-product assumption inherent in the first model. This model acknowledges the multifaceted nature of real-world supply chains, where retailers often need to balance multiple products with varying demand patterns and transportation requirements. Together, the three models offer a thorough examination of the integrated inventory and transportation problem, each introducing greater complexity and closer alignment with real-world scenarios. By progressively relaxing key assumptions and expanding the operational considerations, we provide enhanced insights on the ways retailers can refine their inventory management and transportation policies.

2.In Section 3, in order to facilitate the explanation of the models, I suggest that the authors should organize and display the symbols used in the models in a table and distinguish which ones are parameters and which are decision variables.

Response: We are grateful to the reviewer for this suggestion, which significantly improved the presentation and clarity of our paper. We have included two tables that list the notations for the models' parameters and decision variables, respectively.

3.In Table 3, why are the values of v for all items 0?

Response: We simply assumed that the salvage values of the four products are null. This scenario arises when unsold units of the products, like fruits and vegetables, are discarded at the end of the sales period.

4.In numerical example analysis, is it possible to perform sensitivity analysis of parameters?

Response: We conducted sensitivity analysis to assess the impacts of the most crucial parameter—fixed transportation costs for owned and leased trucks—on the inventory and transportation policies within the first two models. Please see pages 13 and 18.

5.In Section 6, the authors should highlight the academic or theoretical contribution based on the results of the analysis and compare with the results of previous relevant studies to increase the completeness of this study.

Response: We have revised the conclusion section to highlight the academic and theoretical contributions of the paper.

Minor Comments

1.The authors should explain how Figure 1 was drawn. That is, what are the parameter values of these functions?

Response: The figure has been revised by removing some confusing parameters. It now displays only the functions for total expected cost, expected profit, and expected profit specific to the newsvendor problem.

2.Please confirm the number of the form again.

Response: Sorry, it unclear what we need to confirm

3.Page 7, line↑7，”(, )” should be “()”.

Response: Sorry, we couldn’t find the line with “(,)”

4.Page 15, lines 1-6, the solution process for stage 2 is confusing and the authors are advised to state it specifically.

Response: We apologize for the confusion caused by the steps in the second stage of the algorithm. We have reviewed and revised these steps, and hope they are now clear.

Many thanks again for the constructive comments and suggestions

Reviewer #3: 

 Include a brief mention of the results in the abstract.

Response: We have revised the abstract to highlight the results of the paper 

2. Add a table detailing author contributions.

Response: We are grateful to the reviewer for this suggestion, which significantly improved the presentation of our paper. We have included a table that helps to position our paper and highlights its contribution.

3. Number the provided examples.

Response: The illustrative examples are now numbered

4. Ensure that order quantity in Table 4 consists of whole numbers.

Response: The order quantity consists of whole numbers in all tables

5. Expand the conclusion section with detailed explanations and include managerial implications.

Response: We have revised the conclusion to include detailed explanation of the paper contribution and managerial implications. 

6. Include sensitivity analysis of the models.

 Response: We conducted sensitivity analysis to assess the impacts of the most crucial parameter—fixed transportation costs for owned and leased trucks—on the inventory and transportation policies within the first two models. Please see pages 13 and 18.

Many thanks again for the constructive comments and suggestions

---

## [Decision Letter · Decision Letter 1]

20 May 2024

Incorporating transportation costs into the single and multiple items newsvendor problems

PONE-D-24-00923R1

Dear Dr. Moncer,

We’re pleased to inform you that your manuscript has been judged scientifically suitable for publication and will be formally accepted for publication once it meets all outstanding technical requirements.

Kind regards,

Mohammad Heydari

Academic Editor

PLOS ONE

Additional Editor Comments (optional):

The authors did an excellent job, and the paper can be accepted as it is.

Reviewers' comments:

Reviewer's Responses to Questions

**Comments to the Author**

1. If the authors have adequately addressed your comments raised in a previous round of review and you feel that this manuscript is now acceptable for publication, you may indicate that here to bypass the “Comments to the Author” section, enter your conflict of interest statement in the “Confidential to Editor” section, and submit your "Accept" recommendation.

Reviewer #2: All comments have been addressed

Reviewer #4: All comments have been addressed

2. Is the manuscript technically sound, and do the data support the conclusions?

Reviewer #2: Yes

Reviewer #4: Yes

3. Has the statistical analysis been performed appropriately and rigorously? 

Reviewer #2: Yes

Reviewer #4: I Don't Know

4. Have the authors made all data underlying the findings in their manuscript fully available?

Reviewer #2: Yes

Reviewer #4: Yes

5. Is the manuscript presented in an intelligible fashion and written in standard English?

Reviewer #2: Yes

Reviewer #4: Yes

6. Review Comments to the Author

Reviewer #2: The authors have responded to my comments and the present version have been improved significantly compared with the previous version. I suggest that this paper can be accepted for publication when the following comment is replied. First, I found several grammatical errors in writing, please check again. In addition, to make thpaper easier to read, it is suggested that the sections and sub-sections be numbered.

Reviewer #4: (No Response)

7. PLOS authors have the option to publish the peer review history of their article (what does this mean?). If published, this will include your full peer review and any attached files.

Reviewer #2: No

Reviewer #4: No

---

## [Editor Report · Acceptance letter]

5 Jun 2024

PONE-D-24-00923R1 

PLOS ONE

Dear Dr. Hariga, 

I'm pleased to inform you that your manuscript has been deemed suitable for publication in PLOS ONE. Congratulations! Your manuscript is now being handed over to our production team.

Kind regards, 

on behalf of

Scientist Mohammad Heydari 

Academic Editor

PLOS ONE